



# Catchment hydrological response and transport are affected differently by precipitation intensity and antecedent wetness

Julia L.A. Knapp[1,2], Wouter R. Berghuijs[3,2], Marius G. Floriancic[2,4], and James W. Kirchner[2,5]

[1]Department of Earth Sciences, Durham University, Durham, United Kingdom
[2]Department of Environmental Systems Science, ETH Zürich, Zürich, Switzerland
[3]Department of Earth Sciences, Free University Amsterdam, Amsterdam, the Netherlands
[4]Department of Civil, Environmental and Geomatic Engineering, ETH Zürich, Zürich, Switzerland
[5]Swiss Federal Institute for Forest, Snow and Landscape Research (WSL), Birmensdorf, Switzerland

**Correspondence:** Julia L.A. Knapp (julia.l.knapp@durham.ac.uk)

**Abstract.** Hydrological response and transport are distinct catchment behaviours that have both been intensively studied, but rarely together. The hydrologic response characterises how quickly, and how strongly, streamflow reacts to precipitation inputs, whereas transport characterises how quickly precipitation reaches the stream. Here we use sub-daily time series of hydrometeorological fluxes and stable water isotopes to quantify both hydrological response and transport in two intensively studied

temperate catchments. Consistent with previous studies, we find that hydrologic response is much quicker than transport. However, we also find that catchment wetness and precipitation intensity influence hydrologic response and transport in different ways. Increased antecedent wetness results in stronger runoff responses, primarily mobilising more old water, while increased precipitation intensity results in a faster propagation of the runoff response signal, and the delivery of greater proportions of recent precipitation to streamflow. Considered together, response times and travel times provide insights into runoff generation

mechanisms, flow paths, and water sources.

## 1  Introduction

Understanding how catchments store and release water is crucial for accurate streamflow predictions, model development, and characterisation of hydrologic systems. Such understanding also plays a vital role in, e.g., managing water resources effectively (Grathwohl et al., 2013), mitigating flood risks (Peskett et al., 2023), promoting ecosystem health (Laudon and

Sponseller, 2018), and anticipating impacts of climate change (Sulis et al., 2011).

A crucial aspect of catchment water dynamics is the disconnect between the relatively short timescales of hydrologic response to precipitation, and the much longer timescales over which precipitation inputs are transported to the stream. As a consequence of this contrast in time scales, streamflow often responds quickly to precipitation inputs, but this streamflow is typically composed mostly of older waters released from subsurface storage, with only a minor contribution from recent rainfall

(Kirchner, 2003; Neal and Rosier, 1990; Jasechko et al., 2016; Floriancic et al., 2024). Hydrologic response times characterise how quickly precipitation inputs result in rising streamflow, reflecting the celerity with which hydraulic potentials propagate through the catchment. Catchment travel times, by contrast, characterise how quickly precipitation itself reaches the stream,





reflecting the velocity of transport through the catchment and determining the age (i.e., the time since entering the catchment as precipitation) of water leaving the catchment as streamflow (McDonnell and Beven, 2014). Response times or travel times can
be viewed as the path length connecting precipitation to streamflow, divided by the celerity or velocity, respectively. Response and travel times have distributions rather than single values, reflecting the many complex pathways connecting precipitation to streamflow.

Most catchment studies have focused exclusively on either hydrologic response or transport; few have considered both together. For instance, hydrologic response is typically studied in the context of flood behaviour, such as for estimating peak
flows (Gericke and Smithers, 2014) or quantifying flood wave propagation (Meyer et al., 2018). Conversely, tracer-based transport studies aim to understand mixing processes (Botter, 2012; van der Velde et al., 2012; Kirchner et al., 2001, 2000), precipitation partitioning (Soulsby et al., 2011; Botter et al., 2010), and pollutant turnover (Hrachowitz et al., 2016). The few studies that have jointly assessed hydrologic response and transport have consistently found that streamflow responds to rainfall faster than rainwater itself reaches the stream (e.g., van Verseveld et al., 2017; Rasmussen et al., 2000; Seeger and Weiler,
2014; Torres et al., 1998), although in-depth comparisons are rare. The starkly different timescales of streamflow response and transport of water are also neglected in most hydrological, land-surface, and earth system models, which likely leads to flawed representations of water cycling. Previous work considering both response and transport has usually been limited to laboratory column studies and plot-scale or hillslope-scale field studies, due to a lack of sufficient tracer data at the catchment scale. For example, Scaini et al. (2017) assessed the relevance of preferential flow through macropores, concluding that understanding
the "relationship between tracer velocities and wave or wetting front celerity is essential for understanding the complexity of flow" in soils. These findings from small-scale, controlled experiments likely also hold for the larger catchment scale, as is also suggested by the prevalence of fractal scaling in stream chemistry dynamics (Godsey et al., 2010; Kirchner and Neal, 2013; Kirchner et al., 2000). However, a comprehensive joint evaluation of hydrologic response and transport at the catchment scale is missing. Quantifying response and transport processes jointly can provide novel insights into water storage processes and
may also reveal more general emergent functional behaviour and thus aid model development.

Previous work has shown that hydrologic response and transport both depend on catchment properties, ambient conditions, and event characteristics. Hydrologic responses to precipitation inputs are often affected by antecedent catchment wetness (Zehe et al., 2005), with wetter conditions typically resulting in higher event runoff ratios (Schoener and Stone, 2019) and greater event streamflow (Bennett et al., 2018). It has also been suggested that higher antecedent wetness may result in shorter
response times (e.g., Mindham et al., 2023). Likewise, it has been shown that increased antecedent wetness and higher precipitation intensities can promote shorter travel times and thus quicker transport of recent precipitation to the stream (Knapp et al., 2019; Wilusz et al., 2017; Hrachowitz et al., 2009). However, little is known about how strongly antecedent wetness conditions and precipitation intensity affect the relationship between response and transport processes.

In this study, we estimate response and transport metrics using hydrometric and isotope tracer time series from the pre-Alpine
Erlenbach catchment in Switzerland and from the moorland Upper Hafren catchment at Plynlimon, Wales, UK. We hypothesise that catchment wetness and precipitation intensity affect response and transport processes in different ways. We expect that transport is highly dependent on catchment wetness, as increasing catchment wetness and rising groundwater levels result in the





activation of shallower flow paths that deliver larger proportions of recent precipitation to the stream. We expect precipitation

intensity to affect the streamflow response more strongly than transport, because higher-intensity precipitation may lead to

quicker and larger changes in subsurface water potentials, in turn resulting in a faster propagation of hydrologic signals without

changing the underlying processes. We thus hypothesise that hydrologic response and transport metrics provide complementary

information regarding the storage and release of water in catchments, allowing us to further conceptualise processes like time-

variable water release, activation of flow paths, and subsurface water storages. Contrasting hydrologic response and transport

metrics under varying catchment conditions may also help to reveal how the old waters stored in catchments can be released

quickly during precipitation events, providing fundamental insights into catchment functioning.

## 2   Methods

### 2.1   Site descriptions

The Erlenbach is a steep 0.7 km$^2$ catchment with high drainage density in the Swiss Alptal valley (van Meerveld et al., 2018).

Its bedrock is primarily composed of Flysch, which consists of alternating layers of conglomerate and calcareous sandstones

with schists and marlstones (Zobrist et al., 2018). The soils are predominantly Gleysols (Hagedorn et al., 2000; Schleppi et al.,

1998), which have low permeability and are prone to waterlogging (Rinderer et al., 2014). More than half of the catchment

is forested, particularly the steeper sections. Wet meadows cover the flatter parts of the catchment where the water table is

close to the surface (van Meerveld et al., 2018; Stähli et al., 2021). Vegetation in the catchment is dominated by spruce and fir,

and the meadows in the upper parts of the catchment are used for summer grazing. Annual precipitation in the area is around

2300 mm/year (1980-2011), of which around one third falls as snow in the winter months. Mean annual runoff is around 1800

mm/year.

The Upper Hafren is a 1.22 km$^2$ upland catchment at Plynlimon in Mid-Wales, UK. Its bedrock is primarily composed of

Lower Palaeozoic mudstones and shales, which are highly fractured, enabling storage and rapid transport of water (Shand et al.,

2005). The soils in the Upper Hafren catchment are predominantly acidic, organic-rich peats and gleys of low permeability,

typical of upland moorland catchments (Kirby et al., 1991). The catchment's vegetation is dominated by ferns, acidic grassland,

and peaty mires (Hill and Neal, 1997), and the primary land use is sheep grazing. The catchment receives annual precipitation

of approximately 2700 mm/year and the mean annual runoff is around 2400 mm/year (1974-2010).

### 2.2   Description of the data sets

Time series of hydrometeorological data, including precipitation and streamflow, along with stable water isotope tracers (deu-

terium and oxygen-18), were analysed at the Erlenbach and Upper Hafren catchments. To ensure comparability between our

response and transport analyses, we only used hydrometeorological time series spanning the start and end dates of the tracer

measurements.





In the Erlenbach catchment, hydrometeorological time series are available at 10-minute intervals, but are aggregated to 30 min intervals for this analysis. Precipitation is measured at the Erlenhöhe meteorological station (1228 m a.s.l.), while stream-flow is recorded at the Erlenbach outlet (1100 m a.s.l.). Stable water isotopes were measured in precipitation and streamflow at approximately hourly intervals at an in-situ field laboratory located at the Erlenbach outlet (von Freyberg et al., 2017) between 01 August 2016 and 31 July 2020. The isotope time series contain occasional data gaps due to instrumentation malfunctioning, as illustrated in Figure 1a.

In the Upper Hafren catchment at Plynlimon, hydrometeorological data is available at hourly resolution from the Carreg Wen meteorological station (575 m a.s.l.) and the Upper Hafren stream gauge (550 m a.s.l.). Stable water isotopes were measured at 7-hourly intervals in precipitation at Carreg Wen and in streamwater at the Upper Hafren stream gauge from July 2007 through March 2009, but due to a 3-month data gap between December 2007 and mid-March 2008 caused by sample loss, only data from March 2008 until March 2009 were analysed here.

At both sites, streamflow measurements and samples were taken instantaneously, while precipitation measurements and samples are cumulative over the respective sampling interval.

## 2.3 Data Analysis

### 2.3.1 Quantification of antecedent wetness and precipitation intensity

We quantified antecedent wetness by using the stream discharge recorded one time step prior to the isotope sampling and water flux measurement (i.e., one and seven hours earlier, at Erlenbach and Upper Hafren, respectively), consistent with the isotope sampling frequency at each site. Precipitation intensity was calculated as the average precipitation rate during the time interval preceding each streamflow measurement.

### 2.3.2 Splitting criteria

We analysed the time series for different percentile ranges of antecedent wetness and precipitation intensity, to evaluate how these factors affect the hydrologic response and transport in the catchments. When analysing the effect of wetness or precipitation intensity separately, we binned the 0-60th, 60th-80th, 80th-90th, 90th-95th and 95th-100th percentiles at Erlenbach (five data bins), and the 0th-55th, 55th-75th, 75th-90th and 90th-100th percentiles at Upper Hafren (four data bins). The smaller number of data points at Upper Hafren required using larger bin sizes and consequently resulted in fewer bins.

When we split the time series by combinations of antecedent wetness and precipitation intensity, we divide both variables into their 0-60th, 60th-80th, and 80th-100th percentiles at both sites. To focus on the greatest contrasts, we selected all possible pairings of low (0-60th percentile) and high (80th-100th percentile) antecedent wetness and precipitation intensity for this analysis.



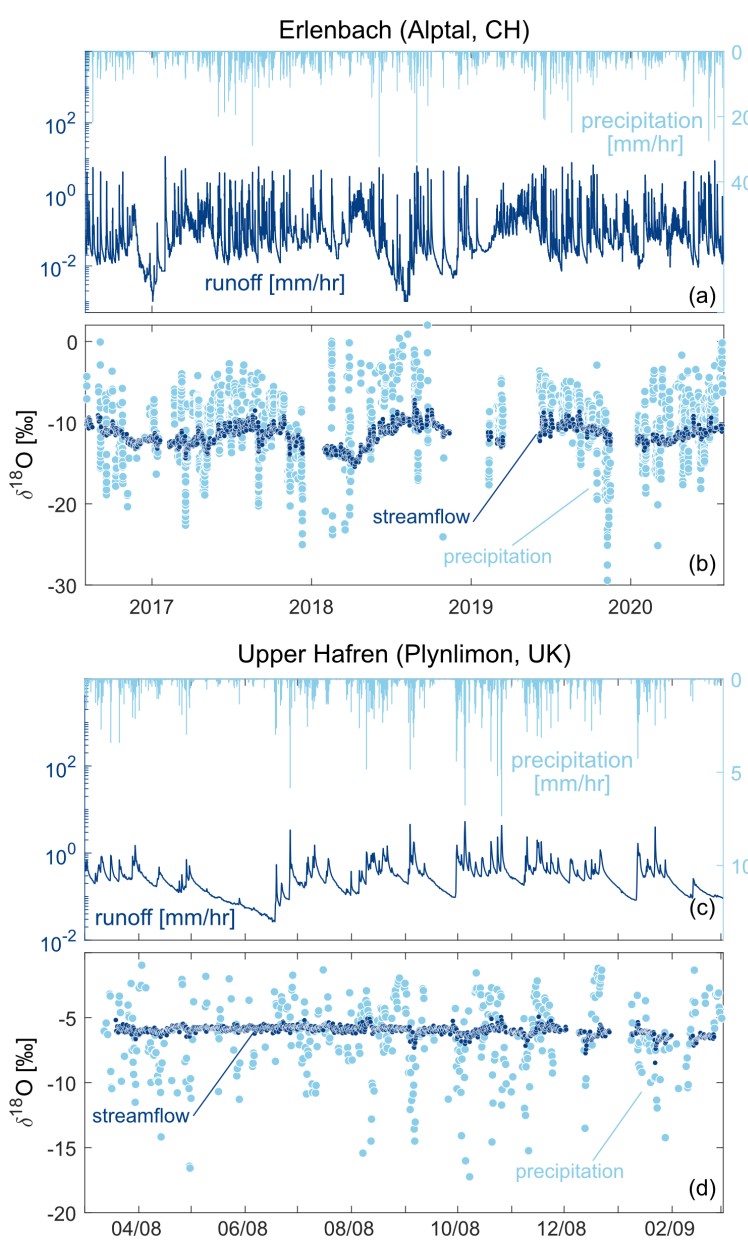

**Figure 1.** Oxygen-18 data in streamflow (dark blue) and precipitation (light blue) as well as streamflow and precipitation water fluxes at Erlenbach (Alptal – a, b) and Upper Hafren (Plynlimon – c, d).

### 2.3.3 Ensemble approaches: Ensemble Rainfall-Runoff Analysis and Ensemble Hydrograph Separation

The hydrometeorological and tracer time series were analysed using ensemble approaches. Unlike classical unit hydrograph analysis methods that examine individual events in isolation, ensemble approaches estimate the characteristic behaviour of col-





lective "ensembles" of events. This allows for the determination of typical hydrologic response and transport processes under specific catchment and event conditions (here, percentiles of antecedent wetness and precipitation intensity).

**Ensemble Rainfall Runoff Analysis** quantifies the distribution of response times and provides insights into how stream discharge responds to precipitation inputs (Kirchner, 2022, 2024a). The runoff response distribution (RRD) quantifies the in-

cremental runoff response per unit precipitation input as a function of lag time (Figure 2 a). The RRD can be converted to the actual streamflow response through multiplication with the precipitation input, yielding the Nonlinear Response Function (Kirchner, 2024a). From the RRD, metrics describing the timing of the peak response ($t_{peak}$), and the height of the peak response ($h_{peak}$), can be derived (Figure 2 c). Both are estimated from a quadratic fit to the uppermost 20% of the RRD. Additionally, the rainfall-runoff coefficient ($C$) is quantified as the integral of the RRD across the range of analysed lag times.


**Ensemble Hydrograph Separation** quantifies the volume-weighted travel time distribution (TTD) and thus transport of water from precipitation to streamflow. The forward TTD determines the relative proportion of precipitation that contributes to streamflow within specific time intervals, while the backward TTD quantifies the proportion of streamwater that consists of water of different ages (Figure 2 b, d). Additional information regarding the approach can be found in Kirchner (2019),

while its application is described in Knapp et al. (2019), and the associated analysis codes are presented in Kirchner and Knapp (2020a). From the TTDs, new water fractions quantifying the amount of water that is "new" since the last sampling can be extracted. We calculated the backward new water fraction for all timesteps with precipitation $^{Q_p}F_{\text{new}}$, and the forward new water fraction $^{P}F_{\text{new}}$. These two metrics are related to each other by:

$$^{P}F_{\text{new}} = {}^{Q_p}F_{\text{new}} \frac{\overline{Q_p}}{\overline{P}} \frac{n_p}{n} \tag{1}$$

where $\overline{Q_p}$ is the average runoff during the $n_p$ time steps with precipitation, and $\overline{P}$ is the average precipitation of all time steps $n$.

Together, these two ensemble analyses can provide insights into the response of streamflow to a unit of recent precipitation and the absolute and relative volumes of recent precipitation found in streamflow.

## 3 Results

In both catchments the hydrologic response was quicker than transport, with higher values of the runoff response distribution (RRD) compared to the travel time distribution (TTD) during the lag times considered here (10 hrs and 48 hrs at Erlenbach and Upper Hafren, respectively, Figure 3). This indicates that much of the quickly mobilised water during storm events consists of older water. On the other hand, the TTDs decayed more slowly than RRDs as functions of increasing lag times, indicating greater persistence in catchment transport than in the hydrological response. These findings illustrate that the runoff response

occurs on much shorter timescales than the actual transport of water through the catchment.

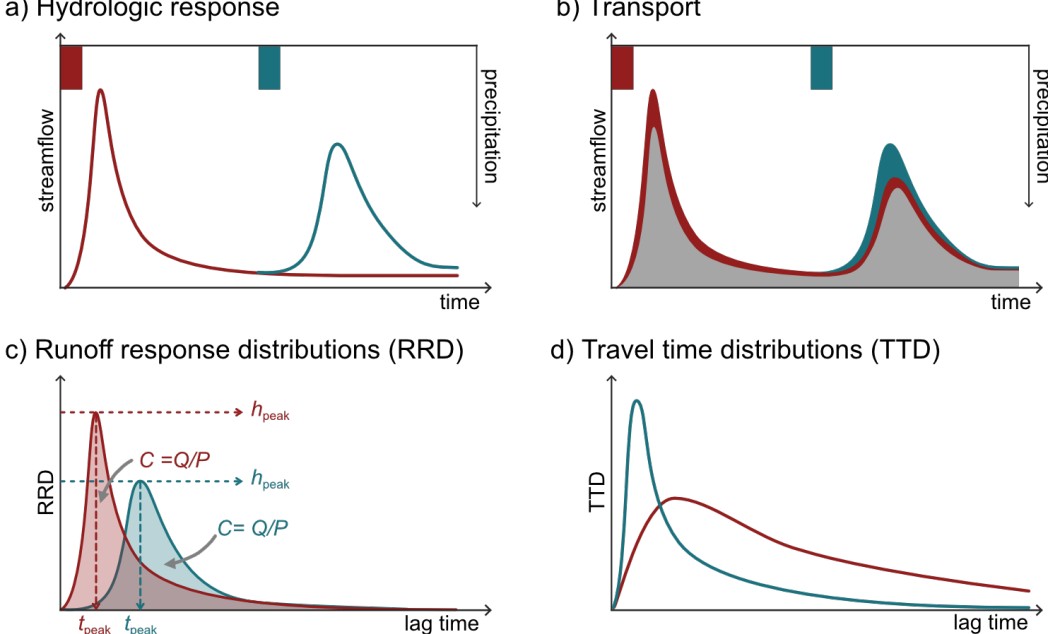

**Figure 2.** Hydrologic response and transport exemplified across two events (red and green). Streamflow is generated in response to precipitation inputs (a), but only some fraction of the generated streamflow consists of precipitation from the most recent precipitation event (b). Runoff response distributions (RRD, c) and travel time distributions (TTD, d) shape the observed response and transport, and distributions can take many different shapes, with ensemble distributions (not shown) characterising the "typical" response under specific conditions. The RRD (c) quantifies the time that the precipitation input takes to generate streamflow, and the illustrated metrics describe runoff characteristics in response to each unit of precipitation (i.e., the peak height $h_{peak}$, the peak time $t_{peak}$ and the runoff coefficient $C$). The TTD (b) quantifies the time for precipitation inputs to become streamflow. New water fractions ($F_{new}$) can be derived from the TTDs and assess the amount of precipitation contributing to streamflow that is "new" since the last sampling of streamflow.

## 3.1 Effect of antecedent wetness

Antecedent wetness affected hydrological response and transport at both the Erlenbach and Upper Hafren catchments (illustrated in blue in Figure 4). The peak height of the RRD $h_{peak}$ and the runoff coefficient $C$ increased with antecedent wetness, more than doubling between dry and wet conditions. This suggests a much greater response in streamflow to the same pre-
cipitation input under wetter conditions (Figure 4 a,c,i,k,m,o), which is similar to the behaviour one would expect from a nonlinear storage-discharge relationship. Notably, the timing of the arrival of the runoff peak $t_{peak}$ did not change substantially with antecedent wetness (Figure 4 e,g), suggesting that the streamflow responses occurred equally quickly during dry and wet conditions.

When tracking transport of water through the catchment, we focused on transport metrics (i.e., the new water fractions
$^{Q_p}F_{\text{new}}$ and $^{P}F_{\text{new}}$) rather than the full TTDs due to the limited number of isotope data points available. Both forward and



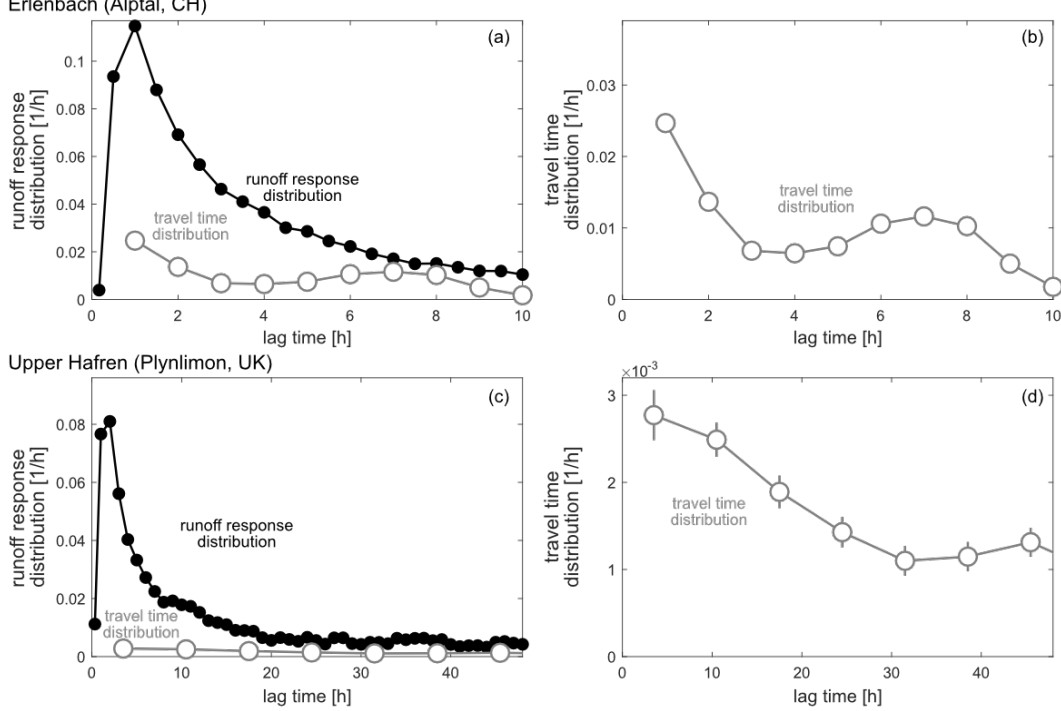

**Figure 3.** A comparison of runoff response distributions (RRDs, black, panels a and c), and backward travel time distributions (TTDs, grey, all panels) at Erlenbach (a, b) and Upper Hafren (c, d). Note that the y-axes of the TTDs are magnified by a factor of 3 for Erlenbach and a factor of 30 for Upper Hafren compared to those of the respective RRDs, because the magnitudes of the TTDs are much smaller than those of the RRDs. This is also illustrated in panels a and c with the TTDs presented in grey next to the RRDs. Error bars representing standard errors are included for all data points; however, in most cases, they are too small to be visually discernible.

backward new water fractions were small (around 5%) and did not increase with greater antecedent wetness (Figure 4 q,s). We also quantified new water fractions over aggregated intervals of 21 hrs (Figure S1). New water fractions were larger for these longer intervals than for the original sampling intervals (1 and 7 hr at Erlenbach and Upper Hafren, respectively), partly as a natural consequence of the fact that the fraction of new water will inherently grow with the interval of water age that is

considered "new" (see section 5.3 of Knapp et al., 2019, for a more detailed explanation). Across all time intervals, however, new water fractions exhibited similar patterns of small increases with antecedent wetness (Figure S1).

Our findings indicate a strong dominance of older water in streamflow and show that antecedent wetness affects the transport of water through the catchment much less than it affects the streamflow response. Intriguingly, the two catchments had similar RRD and TTD metrics, and similar sensitivities to antecedent wetness (Figure 4), despite their substantial differences

in topography, landcover and geology.

**Figure 4.** Effect of changes in antecedent wetness (blue) and precipitation intensity (red) on runoff-response distributions (a-d) and their associated peak times (e-h), peak heights (i-l), runoff coefficients (m-p), and new water fractions (q-t; diamonds indicate forward new water fractions, circles indicate backward new water fractions). Error bars indicate standard errors, where these are larger than the plotting symbols. Both catchments exhibit stronger runoff response under wetter antecedent conditions, but with little change in the amount of recent precipitation reaching the stream. Both catchments also exhibit a stronger and faster response, as well as higher proportions of recent precipitation in streamflow, at higher precipitation intensities.





## 3.2 Effect of precipitation intensity

Precipitation intensity affected the hydrologic response and transport at both catchments (red symbols in Figure 4). Higher precipitation intensities shortened the RRD peak arrival time $t_{peak}$ by factors of approximately 10 at Erlenbach and 4 at Upper Hafren (between lowest and highest precipitation intensities; Figure 4 f,h). Higher precipitation intensities also increased RRD

peak heights $h_{peak}$ (Figure 4 j,l) and runoff coefficients $C$ (Figure 4 n,p), approximately doubling both metrics between the lowest and highest precipitation intensities. These results suggest a stronger and quicker streamflow response to higher-intensity precipitation inputs.

Because RRDs quantify the response per unit of precipitation input, a linear system would yield RRDs that were the same at both high and low precipitation intensities. Thus, our results indicate a nonlinear system, one that reacts faster and more

strongly to higher-intensity precipitation. The magnitude of the actual runoff response, obtained from multiplying the RRD with the precipitation volume (this is the nonlinear response function of Kirchner, 2024a), increases even more strongly with precipitation intensity, as illustrated in Figure 4 j,l (grey symbols).

New water fractions at both sites increased with precipitation intensity, exceeding 10% (i.e., 0.1) for backward new water fractions, while forward new water fractions remained in the range of 2-6% (Figure 4 r,t). This underscores that an increasing

proportion of streamflow originates from recent precipitation as precipitation intensities increase, while overall still only small fractions of recent precipitation reach the stream. New water fractions calculated for 21-hour aggregations were larger than those of the shorter sampling intervals but exhibited similar relationships to precipitation intensity (Figure S1).

Both hydrologic response and transport exhibited similar sensitivity to precipitation intensity at Erlenbach and Upper Hafren, despite their differences in topography, land cover and geology, indicating similar responses of both catchments to variations

in antecedent wetness.

## 3.3 Joint effects of antecedent wetness and precipitation intensity

To evaluate how antecedent wetness and precipitation intensity interact to affect hydrologic response and transport, we compared four scenarios of high/low antecedent wetness and high/low precipitation intensity (Figure 5). Low precipitation intensities combined with dry antecedent conditions resulted in both a weak hydrologic response and negligible transport (pale yellow

symbols in Figure 5). High precipitation intensities combined with wet antecedent conditions resulted in strong and rapid hydrologic response as well as substantial transport (dark purple symbols in Figure 5). The hydrologic response was much weaker for all other combinations of precipitation intensity and antecedent wetness (Figure 5 a,b). However, if the RRD (streamflow response per unit of precipitation) is rescaled by multiplying by the precipitation input (resulting in the nonlinear response function of Kirchner, 2024a), one sees that this response is substantial whenever precipitation intensity is high, regardless of

antecedent wetness (Figure 5 c,d), whereas the response is essentially zero under all scenarios of low precipitation intensity owing to near-zero precipitation.

The combined effect of antecedent wetness and precipitation intensity on transport behaviour differed between forward and backward TTDs. Forward TTDs (Figure 5 e,f) were generally smaller than backward TTDs (Figure 5 g,h), which can





be explained by the fact that they are related to each other by the runoff coefficient (Eq. 1), which is usually smaller than 1.
Forward TTDs at both sites were greatest when both antecedent wetness and precipitation intensity were high, although the
effect was relatively weak. Backward TTDs, on the other hand, appeared to be most affected by precipitation intensity and less
by antecedent wetness, with highest TTDs observed for high precipitation intensity independent of antecedent wetness.

### 3.4 Contrasting and combining response and travel time distributions

The TTD quantifies the proportion of precipitation becoming streamflow (forward TTD) or the proportion of streamflow
consisting of recent precipitation (backward TTD). To quantify the actual amounts of recent precipitation reaching the stream,
we multiplied the forward TTD with the RRD (which quantifies the streamflow generated per unit of rainfall input) and the
total precipitation input (Figure 5 i,j). These calculations illustrate that the only noteworthy signals of recent precipitation
in streamflow occurred under conditions of high precipitation intensity, with the strongest response when high precipitation
intensity was paired with high antecedent wetness (purple symbols in Figure 5 i,j). Wet antecedent conditions combined with
low-intensity precipitation resulted in near-zero amounts of recent precipitation reaching the stream, indicating that wetness
conditions alone only played a minor role in governing transport processes.

### 4 Discussion

Our results reiterate the apparent paradox between catchment hydrologic response and transport, in which streamflow responds
to precipitation inputs almost instantaneously, even though streamflow is primarily composed of old water stored in the catch-
ment (Kirchner, 2003; Małoszewski and Zuber, 1982; Botter et al., 2010; McDonnell, 1990). However, the extent to which
antecedent wetness and precipitation intensity affect streamflow responses and the transmission of recent precipitation to the
stream remains poorly understood. Here we investigated how changes in antecedent wetness and precipitation intensities shape
the hydrologic response and transport across two data-rich catchments. At these two sites, the "old water paradox" consistently
holds true under the full spectrum of precipitation intensities and antecedent wetness conditions.

### 4.1 Conceptualisation of catchment functioning in response to antecedent wetness and precipitation intensity

The interplay between antecedent wetness and precipitation intensity influences hydrologic response and transport, shaping the
overall system behaviour. We found that antecedent wetness and precipitation intensity had different effects on both hydrologic
response and transport. Antecedent wetness primarily influenced the magnitude of the runoff response, while precipitation in-
tensity affected both its magnitude and timing. Notably, only the combination of high precipitation intensity and wet antecedent
conditions led to large runoff responses, which agrees with large-scale studies investigating flood-generating mechanisms (e.g.,
Berghuijs et al., 2019). Transport was also affected differently by antecedent wetness and precipitation intensity: new water
fractions were largely unaffected by changes in wetness but increased with increasing precipitation intensity.

Combining hydrologic response and transport information under various conditions of antecedent wetness and precipitation
intensity can shed light on runoff generation mechanisms and flow paths (Figure 6). Higher catchment wetness resulted in



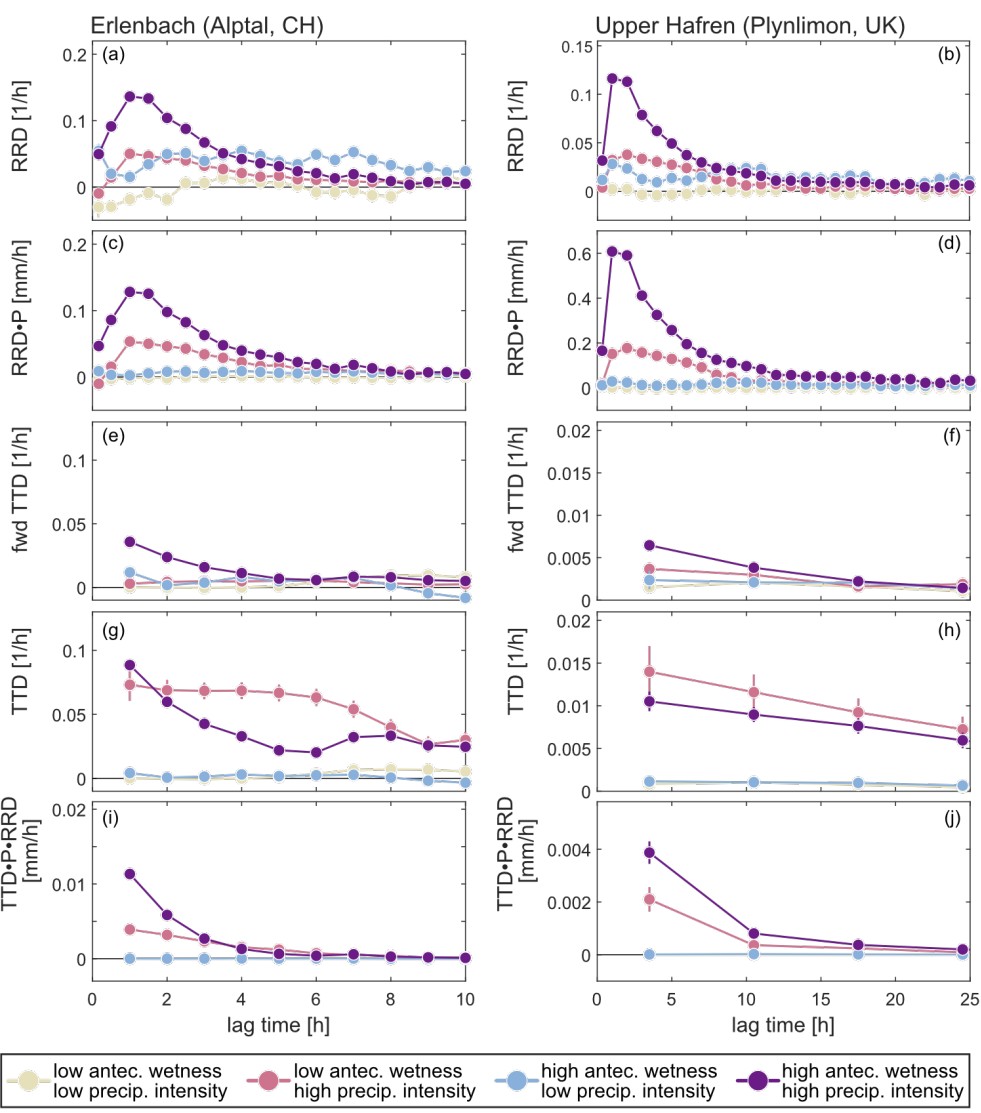

**Figure 5.** Illustration of the combined effect of antecedent wetness and precipitation intensity at Erlenbach (left panels) and Upper Hafren (right panels) on the runoff-response distributions (a, b), the RRD multiplied by the precipitation input (yielding the actual runoff response: c, d), the forward (e, f) and backward (g, h) travel time distributions, and the product of the RRD, TTD and precipitation amount (i, j). Error bars representing standard errors are included for all data points; however, in most cases, they are too small to be visually discernible. The results illustrate that the amount of recent precipitation in streamflow is usually very small, even in cases of strong hydrologic response. Only the combination of high antecedent wetness and high precipitation intensity results in a non-negligible amount of recent precipitation reaching streamflow.

larger hydrologic responses, but the fraction of recent precipitation reaching streamflow did not increase (Figure 6 a,b vs. 6 c,d). Hence, increasing antecedent wetness generates more streamflow by mobilising more old water. This indicates that – contrary



to our original hypothesis – changes in antecedent wetness do not result in greater activation of shallower pathways that deliver more new water to the stream. Instead, increased catchment wetness may increase hydrologic connectivity throughout the catchment, enhancing the connection of stored older waters to the stream, and thus amplifying hydrologic response under wetter conditions. Previous studies highlighted the importance of antecedent wetness in controlling runoff generation (e.g., Penna et al., 2011; James and Roulet, 2007; Detty and McGuire, 2010), but few were able to assess how much recent precipitation is contained in this runoff.

On the other hand, increasing precipitation intensity not only amplifies the runoff response, but also changes the age composition of the water that is being mobilised. This results in a greater (but overall still small) amount of recent precipitation contributing to streamflow at higher precipitation intensities (Figure 6 a,c vs. Figure 6 b,d), as was previously also noted by, e.g., Kirchner (2019) and Knapp et al. (2019). Additionally, the hydrologic response is propagated faster through the system under more intense precipitation, similar to findings by, e.g., Mindham et al. (2023). Our findings regarding the timing and composition of generated runoff indicate that higher precipitation intensity may result in a pronounced activation of shallower flowpaths, ultimately resulting in a stronger and faster streamflow response that is composed of more recent precipitation.

While many studies have investigated hydrologic response and transport separately, the joint representation or hydrologic response and transport provides a more comprehensive picture. Combining information on response and transport allows us to infer the composition of generated runoff under varying conditions of catchment wetness and precipitation intensity. This combined approach also yields potential insights about flowpaths that would not have been achieved by examining hydrologic response and transport in isolation.

## 4.2 The role of wetness and precipitation intensity across catchments

The two catchments investigated in this study have markedly different soil types, topography, and vegetation. Despite these different physical properties, the two catchments exhibited similar patterns in hydrologic response and transport, suggesting that these patterns of behaviour may not be site-specific but may instead be more broadly applicable in diverse landscapes. However, further research is needed to confirm the generality of these findings across a wider range of catchments and environmental conditions. At present, however, these two sites are (to our knowledge) the only ones with sufficiently detailed long-term, high-frequency isotope measurements to facilitate such systematic analyses of the effects of antecedent wetness and precipitation intensity on hydrologic response and transport. Thus, this study highlights the importance of making long-term, high-frequency tracer measurements more widely available.

Many previous studies have related catchment response times to physical properties of the basin (Carrillo et al., 2011; Post and Jakeman, 1996; Nippgen et al., 2011); however, our results demonstrate that the dynamics of runoff response and transport are also strongly dependent on antecedent wetness and precipitation intensity. While catchment-specific characteristics like land use and geology do modulate system behaviour and the activation of flowpaths, our findings underscore the fundamental role of antecedent wetness and precipitation intensity in determining the nature and the dynamics of hydrologic response and transport.





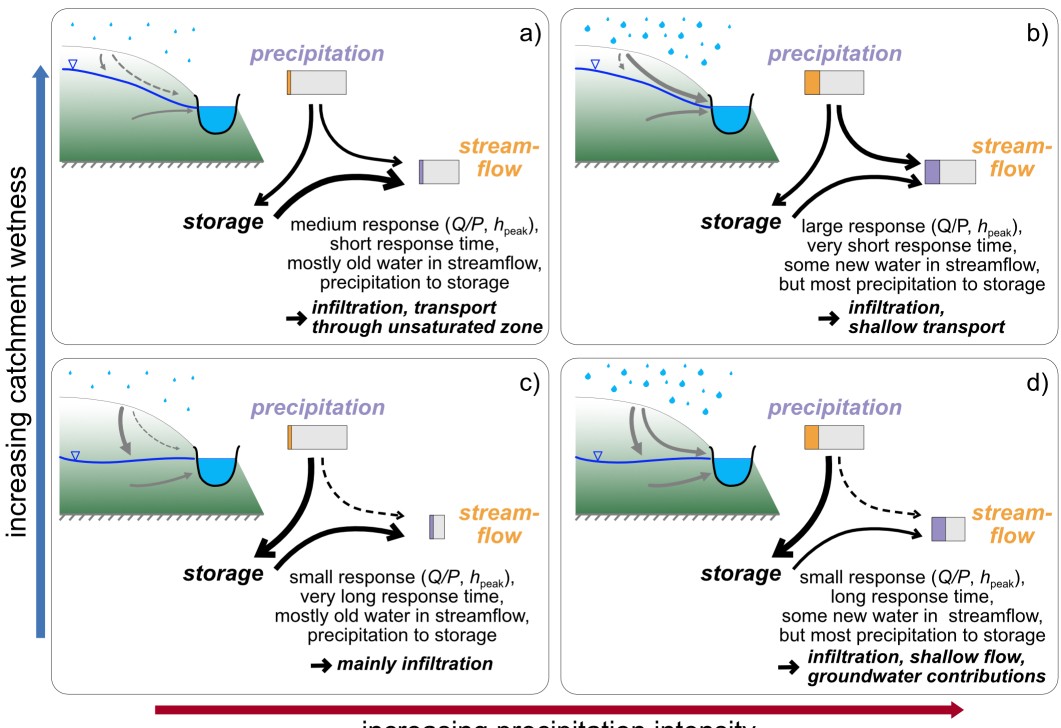

**Figure 6.** Schematic illustration of potential storage and release processes in catchments as functions of high (top) and low (bottom) antecedent wetness and low (left) and high (right) precipitation intensity, as inferred from RRD and TTD metrics. We assume here that larger proportions of recent precipitation in streamflow (i.e., larger new water fractions) imply enhanced transport via shallow subsurface pathways. Larger amounts of older water in streamflow (i.e., smaller new water fractions) are assumed to imply greater contributions from subsurface storage, e.g., groundwater. Widths of arrows indicate the approximate importance of the various pathways, and the relative length of the grey bars for precipitation and streamflow indicate the runoff coefficient (i.e., how much streamflow is produced per unit of precipitation input). The coloured segments of the grey bars indicate how much precipitation promptly becomes streamflow (i.e., forward new water fractions; shown in orange) or which proportion of streamflow is made up of recent precipitation (i.e., backward new water fraction; shown in purple). Per definition, the areas of the orange and purple sections are identical but may represent different proportions of precipitation and streamflow, because they illustrate the same new water from two different perspectives. Wetter antecedent conditions increase the magnitude of streamflow response, but most of this streamflow is composed of older water. Conversely, higher precipitation intensity increases the magnitude of streamflow response, reduces response times, and enables the transport of larger amounts of recent precipitation to the stream.

## 5  Conclusions

Our study shows that antecedent wetness and precipitation intensity have substantial, yet distinctly different, effects on hydrologic response and transport processes. Increases in both antecedent wetness and precipitation intensity result in substantially stronger runoff responses, but through different mechanisms. While higher catchment wetness largely mobilises more old wa-

ter, more intense precipitation mobilises more water of all ages, and increases the proportion of recent precipitation reaching the
stream. Integrating information on hydrologic response and transport thus improves our understanding of relevant catchment
processes.

Our investigation of changes in hydrologic response and transport as functions of antecedent wetness and precipitation
intensity yields a conceptual picture of system behaviour that could potentially be applicable beyond the specific catchments
studied here. However, further research across a wider range of catchments would be required to test this. The effects of changes
in wetness and precipitation intensity on transport are not included in many large-scale models, and their effects on hydrologic
response often remain poorly constrained. Thus, our results reveal key behaviours that could be integrated into models to make
them more realistic.

Many water resource challenges rely not only on the accurate estimation of water fluxes, but also require catchment transport
processes to be captured correctly. Improving our understanding of how different subsurface pathways respond to climatic
shifts or land use changes might enable more accurate and realistic predictions of the effect of wetter catchment conditions
and intensified precipitation on water resources and their quality, with shorter travel times potentially leading to accelerated
pollutant transport (Li et al., 2024). We suggest that joint consideration of hydrologic response and transport processes may
contribute to more accurate water quality and quantity assessments. Such analyses could be relevant for both data-driven
analyses and for model development.

*Code and data availability.*  Isotope and streamflow data from Upper Hafren, Plynlimon are available at https://doi.org/10.16904/envidat.82
(Kirchner et al., 2019). Erlenbach data (isotopes and streamflow records) are available from the authors upon reasonable request. Matlab
and R codes to calculate transport via Ensemble Hydrograph Separation are available at https://doi.org/10.16904/envidat.182 (Kirchner and
Knapp, 2020b). An R code to calculate hydrologic response via Ensemble Rainfall-Runoff Analysis is available at https://www.doi.org/10.
16904/envidat.529 (Kirchner, 2024b).

*Author contributions.*  JLAK and JWK conceived of the study. JLAK analysed the data and wrote the first draft of the manuscript. All authors
contributed to the interpretation of the data and the editing of the manuscript.

*Competing interests.*  At least one of the (co-)authors is a member of the editorial board of Hydrology and Earth System Sciences.

*Acknowledgements.*  The authors received no specific funding for this work.



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
