# Peer review of "Catchment hydrological response and transport are affected differently by precipitation intensity and antecedent wetness"

_Hydrology and Earth System Sciences, 2024_

## Referee Comment (RC1)

In this manuscript, the authors execute a very well-designed, straightforward experiment to isolate the individual effects of antecedent wetness and precipitation intensity on both, response and travel times. They further provide a very clear, systematic, complete description of their results and the associated implications. I really like the intriguing simplicity of the research question: in a hindsight it is such an obvious question. Yet, nobody or very few(?) have explicitly addressed it before. It was a pleasure to read the manuscript and I would be glad to see it published soon.

I have only a few very minor comments and suggestions:

(1) P.1, l.1: the opening sentence of the abstract may put the reader on the wrong footing. I would argue that there are quite a lot of studies that – implicitly or explicitly – analyse hydrological response and transport together: starting from early combined descriptions in models (e.g. Niemi, 1977; Christophersen and Wright, 1981; Bergström et al., 1985) to the surge of studies using coupled hydrological-tracer models over the past decade (e.g. Birkel et al., 2010,2016; Fenicia et al., 2010; McMillan et al., 2012; Hrachowitz et al., 2013, 2021; Harman, 2015, 2019; Benettin et al., 2015, 2017 and many others).
It only becomes clear after reading the paper, what the authors meant to express in this first line. I think it would be helpful for the reader if this statement was rephrased so as to more accurately reflect what has rarely been analysed together and compared with each other: response and transit times.

(2) P.1, l.2-3: again a bit ambiguous and not entirely clear. I recommend to rephrase to also allow readers who are not yet in detail familiar with the issue to understand the meaning and difference between "[...] how strongly, streamflow reacts to precipitation inputs [...]" and "[...] how quickly precipitation reaches the stream."

(3) P.1 l.16: would be good to perhaps use a different word than "crucial" here as it has already been used four lines above.

(4) P.1, l.22: see also comment (2). The statement "[...] how quickly precipitation itself reaches the stream [...]" leaves quite room for ambiguity and could benefit from a more precise formulation.

(5) P.2, l.33-35: I think the paper by Weiler et al. (2003) as one of the earlier studies that made an *explicit* difference between response and travel time distributions needs to be cited here, too.

(6) P.2 l.35-37: Related to comment (1) above, I only partly agree and believe that at least some references to coupled hydrological-tracer models should be mentioned here.

(7) P.5, Fig.1: it is of course not the intention of the manuscript to compare the actual hydrological/tracer response dynamics of the two catchments. However, using the same y-axes scales would still help the reader to easier understand differences between the catchments. Although, for readability of figures, this may not be possible everywhere, here panels (a) and (c) but also panels (b) and (d) could easily have matching y-axes scales.

(8) P.6, l.146 and Fig.3: not clear where the difference in considered lag times comes from. How was this decided and why? In addition, it would be good to use the same y-axes scales for panels (a) and (c).

(9) P.9, Fig.4: without any loss of relevant information, the y-axes scales in each row, i.e. panels (a)-(d); (e)-(h); etc. can be matched. It would make the figure less noisy while also giving the reader a (little) bit more information.

(10) P.14, Fig.6 (see below): not clear why infiltration (red circle symbols) is lower at higher wetness? Is this meant to be a consequence of reduced infiltration capacity? If yes, then should the shallow subsurface infiltration/drainage (blue circles) not also be reduced as water also initially needs to infiltrate to reach these shallow drainage flow paths?! In addition, should infiltration then not also be reduced with increasing precipitation intensity?

[Figure]

Best regards,

Markus Hrachowitz

References:

Benettin, P., Bailey, S. W., Campbell, J. L., Green, M. B., Rinaldo, A., Likens, G. E., ... & Botter, G. (2015). Linking water age and solute dynamics in streamflow at the Hubbard Brook Experimental Forest, NH, USA. Water Resources Research, 51(11), 9256-9272.

Benettin, P., Soulsby, C., Birkel, C., Tetzlaff, D., Botter, G., & Rinaldo, A. (2017). Using SAS functions and high-resolution isotope data to unravel travel time distributions in headwater catchments. Water Resources Research, 53(3), 1864-1878.

Bergström, S., Carlsson, B., Sandberg, G., & Maxe, L. (1985). Integrated modelling of runoff, alkalinity, and pH on a daily basis. Hydrology Research, 16(2), 89-104.

Birkel, C., Dunn, S. M., Tetzlaff, D., & Soulsby, C. (2010). Assessing the value of high-resolution isotope tracer data in the stepwise development of a lumped conceptual rainfall–runoff model. Hydrological Processes, 24(16), 2335-2348.

Birkel, C., Geris, J., Molina, M. J., Mendez, C., Arce, R., Dick, J., ... & Soulsby, C. (2016). Hydroclimatic controls on non-stationary stream water ages in humid tropical catchments. Journal of Hydrology, 542, 231-240.

Christophersen, N., & Wright, R. F. (1981). Sulfate budget and a model for sulfate concentrations in stream water at Birkenes, a small forested catchment in southernmost Norway. Water Resources Research, 17(2), 377-389.

Fenicia, F., Wrede, S., Kavetski, D., Pfister, L., Hoffmann, L., Savenije, H. H., & McDonnell, J. J. (2010). Assessing the impact of mixing assumptions on the estimation of streamwater mean residence time. Hydrological Processes, 24(12), 1730-1741.

Harman, C. J. (2015). Time-variable transit time distributions and transport: Theory and application to storage-dependent transport of chloride in a watershed. Water Resources Research, 51(1), 1-30.

Harman, C. J. (2019). Age-ranked storage-discharge relations: A unified description of spatially lumped flow and water age in hydrologic systems. Water Resources Research, 55(8), 7143-7165.

Hrachowitz, M., Savenije, H., Bogaard, T. A., Tetzlaff, D., & Soulsby, C. (2013). What can flux tracking teach us about water age distribution patterns and their temporal dynamics?. Hydrology and Earth System Sciences, 17(2), 533-564.

Hrachowitz, M., Stockinger, M., Coenders-Gerrits, M., van der Ent, R., Bogena, H., Lücke, A., & Stumpp, C. (2021). Reduction of vegetation-accessible water storage capacity after deforestation affects catchment travel time distributions and increases young water fractions in a headwater catchment. Hydrology and Earth System Sciences, 25(9), 4887-4915.

McMillan, H., Tetzlaff, D., Clark, M., & Soulsby, C. (2012). Do time-variable tracers aid the evaluation of hydrological model structure? A multimodel approach. Water Resources Research, 48(5).

Niemi, A. J. (1977). Residence time distributions of variable flow processes. The International Journal of Applied Radiation and Isotopes, 28(10-11), 855-860.

Weiler, M., McGlynn, B. L., McGuire, K. J., & McDonnell, J. J. (2003). How does rainfall become runoff? A combined tracer and runoff transfer function approach. Water Resources Research, 39(11).

---

## Author Comment (AC1)

Please find below our responses to the comments by Markus Hrachowitz. Our responses are given in blue, and text that will be added during the revision is underlined.

In this manuscript, the authors execute a very well-designed, straightforward experiment to isolate the individual effects of antecedent wetness and precipitation intensity on both, response and travel times. They further provide a very clear, systematic, complete description of their results and the associated implications. I really like the intriguing simplicity of the research question: in a hindsight it is such an obvious question. Yet, nobody or very few(?) have explicitly addressed it before. It was a pleasure to read the manuscript and I would be glad to see it published soon.

We are grateful to Markus Hrachowitz for taking the time to read the manuscript and for the positive evaluation of our work.

I have only a few very minor comments and suggestions:

(1) P.1, l.1: the opening sentence of the abstract may put the reader on the wrong footing. I would argue that there are quite a lot of studies that – implicitly or explicitly – analyse hydrological response and transport together: starting from early combined descriptions in models (e.g. Niemi, 1977; Christophersen and Wright, 1981; Bergström et al., 1985) to the surge of studies using coupled hydrological-tracer models over the past decade (e.g. Birkel et al., 2010,2016; Fenicia et al., 2010; McMillan et al., 2012; Hrachowitz et al., 2013, 2021; Harman, 2015, 2019; Benettin et al., 2015, 2017 and many others).

It only becomes clear after reading the paper, what the authors meant to express in this first line. I think it would be helpful for the reader if this statement was rephrased so as to more accurately reflect what has rarely been analysed together and compared with each other: response and transit times.

We agree. We will reformulate the first sentence of the abstract as follows "Hydrological response and _travel times characterise_ distinct catchment behaviours that have both been intensively studied, but rarely together".

(2) P.1, l.2-3: again a bit ambiguous and not entirely clear. I recommend to rephrase to also allow readers who are not yet in detail familiar with the issue to understand the meaning and difference between "[...] how strongly, streamflow reacts to precipitation inputs [...]" and "[...] how quickly precipitation reaches the stream."

We agree. We will change this second sentence of the abstract to "The hydrologic response characterises how quickly, and how strongly, streamflow reacts to precipitation inputs, whereas transport characterises how quickly precipitation _travels through the system to reach_ the stream".

(3) P.1 l.16: would be good to perhaps use a different word than "crucial" here as it has already been used four lines above.

Agree. We will change "crucial" to "*essential*" here.

(4) P.1, l.22: see also comment (2). The statement "[…] how quickly precipitation itself reaches the stream […]" leaves quite room for ambiguity and could benefit from a more precise formulation.

Thank you. We will change this sentence to: "Catchment travel times, by contrast, characterise how quickly precipitation itself *travels through landscapes to* reach the stream"

(5) P.2, l.33-35: I think the paper by Weiler et al. (2003) as one of the earlier studies that made an *explicit* difference between response and travel time distributions needs to be cited here, too.

Thank you. We will add a reference to Weiler et al. (2003) here.

(6) P.2 l.35-37: Related to comment (1) above, I only partly agree and believe that at least some references to coupled hydrological-tracer models should be mentioned here.

Thank you for this remark. We will edit the text and include some of the references: "The starkly different timescales of streamflow response and transport of water are also neglected in most hydrological, land-surface, and earth system models, which likely leads to flawed representations of water cycling. *Some coupled analyses of hydrological response and transport estimated from tracer data exist (e.g. Hrachowitz et al., 2013; Birkel et al., 2016; Harman, 2015), however,* most work *explicitly comparing* response and transport *timescales* has been limited to laboratory column studies and plot-scale or hillslope-scale *modelling* studies, due to a lack of sufficient tracer data at the catchment scale."

(7) P.5, Fig.1: it is of course not the intention of the manuscript to compare the actual hydrological/tracer response dynamics of the two catchments. However, using the same y-axes scales would still help the reader to easier understand differences between the catchments. Although, for readability of figures, this may not be possible everywhere, here panels (a) and (c) but also panels (b) and (d) could easily have matching y-axes scales.

We respectfully disagree. As noted by the reviewer, it is not the intention to compare the precipitation and streamflow water fluxes and isotope signals between the two sites. On the contrary, there are several reasons to expect that the fluxes and signals differ

between the two catchments (due to differences in climate, land cover, and sampling frequency).

What is of relevance for the analysis is the comparison of P and Q dynamics of water fluxes and isotope signals within the specific sites. If we matched the y-axes of the different sites, the variability within the individual catchments would become less visible, obscuring the dynamics, in particular at Plynlimon/Upper Hafren. Hence, we argue that – for this figure – keeping the y-axes scales different is important for clearly showing the variability in each catchment's data.

(8) P.6, l.146 and Fig.3: not clear where the difference in considered lag times comes from. How was this decided and why? In addition, it would be good to use the same y-axes scales for panels (a) and (c).

Thank you. We assume the reviewer is referring to the differences in lag times between Erlenbach (up to 10 hrs) and Upper Hafren (up to 50 hrs). This is due to the Erlenbach being much flashier than the Upper Hafren. We will update the beginning of the results section to explain this: "In both catchments the hydrologic response was quicker than transport, with higher values of the runoff response distribution (RRD) compared to the travel time distribution (TTD) during the lag times considered here (10 hrs and 48 hrs at Erlenbach and Upper Hafren, respectively, Figure 3). *The shorter lag times at Erlenbach indicate a flashier system compared to the Upper Hafren, where the hydrologic response and transport are much slower.*"

We will update Fig. 3 to ensure the same y-axes scales for panels (a) and (c):

[Figure]

(9) P.9, Fig.4: without any loss of relevant information, the y-axes scales in each row, i.e. panels (a)-(d); (e)-(h); etc. can be matched. It would make the figure less noisy while also giving the reader a (little) bit more information.

Thank you. We will update Fig. 4 to show matching y-axes as per below:

[Figure]

(10) P.14, Fig.6: not clear why infiltration (red circle symbols) is lower at higher wetness? Is this meant to be a consequence of reduced infiltration capacity? If yes, then should the shallow subsurface infiltration/drainage (blue circles) not also be reduced as water also initially needs to infiltrate to reach these shallow drainage flow paths?! In addition, should infiltration then not also be reduced with increasing precipitation intensity?

We agree. We have updated the arrows in the panels to indicate that (1) with increasing antecedent wetness more old water (presumably groundwater) is mobilised – with no differences in infiltration between drier and wetter conditions; and that (2) increased precipitation intensity results in greater transport of recent precipitation to the stream (presumably through shallow flowpaths).

[Figure]

References to be added during the revisions:

Birkel, C., Geris, J., Molina, M. J., Mendez, C., Arce, R., Dick, J., ... & Soulsby, C. (2016). Hydroclimatic controls on non-stationary stream water ages in humid tropical catchments. Journal of Hydrology, 542, 231-240.

Harman, C. J. (2015). Time-variable transit time distributions and transport: Theory and application to storage-dependent transport of chloride in a watershed. Water Resources Research, 51(1), 1-30.

Hrachowitz, M., Savenije, H., Bogaard, T. A., Tetzlaff, D., & Soulsby, C. (2013). What can flux tracking teach us about water age distribution patterns and their temporal dynamics?. Hydrology and Earth System Sciences, 17(2), 533-564.

Weiler, M., McGlynn, B. L., McGuire, K. J., & McDonnell, J. J. (2003). How does rainfall become runoff? A combined tracer and runoff transfer function approach. Water Resources Research, 39(11).

References:

Benettin, P., Bailey, S. W., Campbell, J. L., Green, M. B., Rinaldo, A., Likens, G. E., ... & Botter, G. (2015). Linking water age and solute dynamics in streamflow at the Hubbard Brook Experimental Forest, NH, USA. Water Resources Research, 51(11), 9256-9272.

Benettin, P., Soulsby, C., Birkel, C., Tetzlaff, D., Botter, G., & Rinaldo, A. (2017). Using SAS functions and high-resolution isotope data to unravel travel time distributions in headwater catchments. Water Resources Research, 53(3), 1864-1878.

Bergström, S., Carlsson, B., Sandberg, G., & Maxe, L. (1985). Integrated modelling of runoff, alkalinity, and pH on a daily basis. Hydrology Research, 16(2), 89-104.

Birkel, C., Dunn, S. M., Tetzlaff, D., & Soulsby, C. (2010). Assessing the value of high-resolution isotope tracer data in the stepwise development of a lumped conceptual rainfall–runoff model. Hydrological Processes, 24(16), 2335-2348.

Birkel, C., Geris, J., Molina, M. J., Mendez, C., Arce, R., Dick, J., ... & Soulsby, C. (2016). Hydroclimatic controls on non-stationary stream water ages in humid tropical catchments. Journal of Hydrology, 542, 231-240.

Christophersen, N., & Wright, R. F. (1981). Sulfate budget and a model for sulfate concentrations in stream water at Birkenes, a small forested catchment in southernmost Norway. Water Resources Research, 17(2), 377-389.

Fenicia, F., Wrede, S., Kavetski, D., Pfister, L., Hoffmann, L., Savenije, H. H., & McDonnell, J. J. (2010). Assessing the impact of mixing assumptions on the estimation of streamwater mean residence time. Hydrological Processes, 24(12), 1730-1741.

Harman, C. J. (2015). Time-variable transit time distributions and transport: Theory and application to storage-dependent transport of chloride in a watershed. Water Resources Research, 51(1), 1-30.

Harman, C. J. (2019). Age-ranked storage-discharge relations: A unified description of spatially lumped flow and water age in hydrologic systems. Water Resources Research, 55(8), 7143-7165.

Hrachowitz, M., Savenije, H., Bogaard, T. A., Tetzlaff, D., & Soulsby, C. (2013). What can flux tracking teach us about water age distribution patterns and their temporal dynamics?. Hydrology and Earth System Sciences, 17(2), 533-564.

Hrachowitz, M., Stockinger, M., Coenders-Gerrits, M., van der Ent, R., Bogena, H., Lücke, A., & Stumpp, C. (2021). Reduction of vegetation-accessible water storage capacity after deforestation affects catchment travel time distributions and increases young water fractions in a headwater catchment. Hydrology and Earth System Sciences, 25(9), 4887-4915.

McMillan, H., Tetzlaff, D., Clark, M., & Soulsby, C. (2012). Do time-variable tracers aid the evaluation of hydrological model structure? A multimodel approach. Water Resources Research, 48(5).

Niemi, A. J. (1977). Residence time distributions of variable flow processes. The International Journal of Applied Radiation and Isotopes, 28(10-11), 855-860.

Weiler, M., McGlynn, B. L., McGuire, K. J., & McDonnell, J. J. (2003). How does rainfall become runoff? A combined tracer and runoff transfer function approach. Water Resources Research, 39(11).

---

## Author Comment (AC2)

Please find below our responses to the comments by the anonymous reviewer RC2. Our responses are given in blue, and text that will be added during the revision is underlined.

The study by Knapp et al. investigates how precipitation intensity and antecedent wetness differently affect catchment hydrological response and transport. Using sub-daily hydrometeorological data and stable water isotopes in two catchments in Europe, the authors give evidence that hydrological response (streamflow reaction to rainfall) is much faster than transport (movement of precipitation to the stream). They find that increased antecedent wetness strengthens runoff responses by mobilizing older water, whereas higher precipitation intensity accelerates runoff signals and delivers more recent precipitation to streamflow. These findings provide insights into runoff generation, flow paths, and water sources.

The paper is written well and concise. The applied methodology and its findings are novel in will certainly contribute in advance catchment research. Some suggestions for minor revision in the following.

We are grateful to RC2 for the positive evaluation of our work.

Introduction:

I think the "old water paradox " needs to be presented in a bit more detail. It is indeed not as detailed as what this study is about to present but it is worth mentioning that it has been known for a very long time that hydrodynamic responses have been found to be much faster than hydrogeochemical responses (by different tracers, often using end-member mixing analysis or similar)

As I remember the discussion of some of these studies, there were more possible explanations for this behavior. Wetness, but also something called transmissivity feedback, and the possibility, that unsaturated (but almost saturated) zones become saturated and all the "old water" that was previously immobile, becomes mobile again, e.g. in the riparian zone. Maybe these explanations could also be mentioned and possibly discussed.

Agree. We will add the following text to the introduction to further explain the old water paradox and the differences in hydrodynamic and hydrochemical response timescales: *"This "old water paradox" has been recognised for several decades (e.g. Małoszewski and Zuber, 1982; Rodhe et al., 1996; Weiler et al., 2003) and describes the discrepancy between the fast hydrological response to precipitation and the much slower hydrogeochemical response, indicating that streamflow is primarily composed of older subsurface water rather than recent rainfall."*

Site descriptions and datasets

These two subsections are relatively short, which makes sense. But since differences of the two sites will be discussed later, it would be helpful seeing a direct comparison of their main characteristics in a table

We politely disagree. Relevant site characteristics are included in the text. The main findings of our research suggest that the two sites exhibit broadly similar behaviour, i.e. that the behaviour is potentially independent of catchment characteristics. Hence, we think that adding a table which emphasises catchment characteristics would distract from this general behaviour and could falsely create the impression that this is about site-specific behaviour instead.

Data analysis

Wetness quantification: it seems that wetness is considered to be related to the mobile storage, which is considered to be in relations with discharge. This would disregard wetness in the soils or the unsaturated zone, right? If so, please clarify. I think this might also be a possible reason why changes in antecedent wetness do not have an apparent effect on shallow flow paths. It might, if you quantify it differently (e.f. subsurface storm flow doesn't require saturated conditions, so the wetness metric of this study wouldn't be a good proxy for it)

We consider antecedent discharge as a measure of wetness. While we acknowledge that this does not directly capture soil wetness in the unsaturated zone, we deliberately chose this metric because antecedent discharge provides an integrated measure of catchment-wide wetness. In contrast, soil moisture measurements are often spatially limited to specific locations and may not fully represent the overall wetness state of the catchment. To clarify this point, we will add the following sentence to the manuscript: "*This approach of assessing antecedent wetness ensures that our wetness metric provides a more holistic representation of catchment-wide wetness compared to localised soil moisture variations*."

Figure 1: if space allows it, it would be good seeing both subfigures beside each other (discharge/P and isotopes of both catchments besides each other)

Agree. We will change this as suggested:

[Figure]

Subsection 2.3.3: in addition to the equation some explanation should be give on why backward and forward new water fractions are calculated and how they can be interpreted. This is well done in 3.4 but could be mentioned already up here.

Agree. Section 2.3.3 already contains the following explanation of the backward and forward TTDs: "The forward TTD determines the relative proportion of precipitation that contributes to streamflow within specific time intervals, while the backward TTD quantifies the proportion of streamwater that consists of water of different ages (Figure 2 b, d)." We will add the following: "*Their definition corresponds to the definitions of the backward and forward TTDs, with backward new water fractions quantifying the relative amount of streamflow that consists of precipitation that is "new" during the interval since the previous water sample, and forward new water fractions quantifying the proportion of precipitation that became streamflow during the same interval.*"

Results and discussion

I think it would be worth clarifying what the term "infiltration" is meaning in this study as it seems that it is used to describe infiltration into the groundwater storage/aquifer, which is often referred to by "recharge". I assume that the authors had good reasons to speak about infiltration rather than recharge (or storage, rather than aquifer) but it would be good mentioning this early in the manuscript.

We use the term infiltration rather than recharge in Fig 6 because our focus is on the precipitation perspective rather than groundwater storage (i.e. "where does precipitation

go?" rather than "how are subsurface water volumes changing?"). The term recharge is more commonly associated with groundwater (storage) dynamics and refers to water reaching the saturated zone, which we do not explicitly assess here.

I very much like figure 6! But it also made me think how much it adds to the knowledge compared to the pre-event water / event water studies that have been published in the past. The methodology and datasets of this study are new and unique. In the results section the authors reveal very differentiated insights over hydrodynamics and transport dynamics over a large number of events. I think, if this differentiated information could be somehow added to this figure or be added to the general take-home message of the paper, it would be even more appealing to read it.

We thank RC2 for the positive evaluation of our figure 6. We appreciate the point about how our study builds on and extends previous work. We will modify and clarify the take-home message by emphasising the detailed insights our results provide on hydrologic response and transport dynamics by adding the following at the start of the discussion: *"This study builds on previous research by providing a more detailed view of hydrological and transport dynamics across a large number of events. Our findings highlight how catchment wetness, flow pathways, and solute transport vary under different conditions, offering a more nuanced understanding of catchment responses."*

Finally, in a volume not too excessive, it would also be great to discuss the spatial domain to which the perceptual model in figure 6 could be applied when thinking beyond hillslope scales, and to which types of aquifer it may be most applicable to. Especially in flat terrains and for larger catchments, only regions closer to the stream show such obvious behaviour. I am wondering about it as it brings up the question how well groundwater models and surface models should be coupled and whether lumped or distributed schemes ar most adequate to accommodate the observed processes.

This is an interesting idea and we thank the reviewer for raising this comment. However, we think this is beyond the scope of this study and goes beyond the interpretability of the dataset. We therefore only add the following sentence to the Discussion: "*The spatial scales of these processes may depend on catchment characteristics, such as slope and topography*".